# Electronic and Thermal Properties of Graphene and Recent Advances in Graphene Based Electronics Applications

**DOI:** 10.3390/nano9030374

**Published:** 2019-03-05

**Authors:** Mingyu Sang, Jongwoon Shin, Kiho Kim, Ki Jun Yu

**Affiliations:** School of Electrical & Electronic Engineering, Yonsei University, Seoul 03722, Korea; sangmg315@yonsei.ac.kr (M.S.); shinjw6435@yonsei.ac.kr (J.S.); rlarlgh03@naver.com (K.K.)

**Keywords:** graphene, electronic and thermal properties, electronic and thermal conductivity, quantum Hall effect, Dirac fermions, Seebeck coefficient, thermoelectric effect, graphene-based applications

## Abstract

Recently, graphene has been extensively researched in fundamental science and engineering fields and has been developed for various electronic applications in emerging technologies owing to its outstanding material properties, including superior electronic, thermal, optical and mechanical properties. Thus, graphene has enabled substantial progress in the development of the current electronic systems. Here, we introduce the most important electronic and thermal properties of graphene, including its high conductivity, quantum Hall effect, Dirac fermions, high Seebeck coefficient and thermoelectric effects. We also present up-to-date graphene-based applications: optical devices, electronic and thermal sensors, and energy management systems. These applications pave the way for advanced biomedical engineering, reliable human therapy, and environmental protection. In this review, we show that the development of graphene suggests substantial improvements in current electronic technologies and applications in healthcare systems.

## 1. Introduction

Graphene is a recently discovered two-dimensional (2D) carbon allotrope that consists of only a single layer of carbon atoms arranged in a honeycomb lattice and is a base unit for other graphitic materials. Although graphene has been theoretically studied for several decades [1,2,3], freestanding graphene was first obtained in 2004 by using sticky tape and a pencil, and follow-up experiments opened ‘the world of graphene’ [4,5,6,7]. Over the past decades, various electronics, such as next-generation radio-frequency and high-speed devices, sensors, thermally and electrically conductive composites, and transparent electrodes for solar cells and displays, have been widely developed [8]. Additionally, novel materials such as quantum dots (QDs) and rare earth elements have been thoroughly studied to support the fast-growing technology and to continuously enhance the performance of electronics [9,10,11]. However, the difficulty of synthesis and high cost due to limited supply have made the broad exploitation of these materials difficult [12]. Graphene is an excellent candidate to replace these materials. Graphene exhibits remarkable electronic and thermal properties and shows unusual electronic properties, such as Dirac fermions [13], the quantum Hall effect (QHE) [7], and an ambipolar electric field effect [14]. Low-energy excitations of graphene are massless; Dirac fermions behave uniquely in magnetic fields and lead to the QHE. Additionally, graphene is highly thermally conductive, exhibiting a thermal conductivity of ~4000 Wm^−1^ K^−1^ [15,16,17]. Moreover, graphene has a high Seebeck coefficient and figure of merit, which make it easier to convert electrical current to heat [18]. Thus, graphene has the potential for use in energy-harvesting applications. Additionally, this unique material has impressive mechanical and optical properties, such as a fracture strength of 130 GPa [19], optical transparency [20], high room-temperature carrier mobility, and an ultrabroad optical absorption spectrum [21]. Graphene has attracted explosive interest in physics, material science and wide technological applications [22,23,24,25,26,27]. Here, we present a review of the electronic and thermal properties of graphene and its up-to-date applications, including high conductivity, the quantum Hall effect, Dirac fermions, a high Seebeck coefficient, thermoelectric effects, optical devices, electronic and thermal sensors, and energy management systems. In addition, the remarkable potential to extend the field of applications based on graphene is suggested. 

## 2. Electronic Properties and Applications

### 2.1. Electronic Properties

The remarkable electronic and optical properties observed for graphene crystallites are the primary reasons for the exclusive focus of experimental and theoretical efforts on graphene while ignoring the existence of other 2D materials. The electrons in graphene have long mean free paths without disrupting the electron-electron interactions and disorder. Therefore, the properties of graphene differ from those of other common metals and semiconductors associated with the physical structures and electronic properties. 

#### 2.1.1. Honeycomb Lattice and Brillouin Zone

Figure 1a shows the hexagonal structure of carbon atoms in graphene [13]. A triangular lattice with a basis of two atoms forms the honeycomb lattice structure. The unit vectors of the lattice can be expressed as
(1)$$a_{1}=\frac{a}{2}\left(3,\sqrt{3}\right),\;a_{2}=\frac{a}{2}\left(3,-\sqrt{3}\right),$$
where the distance between two carbon atoms is approximately 1.42 Å. The reciprocal-lattice vectors can be written as
(2)$$b_{1}=\frac{2\pi }{3a}\left(1,\sqrt{3}\right),\;b_{2}=\frac{2\pi }{3a}\left(1,-\sqrt{3}\right).$$
The two points *K* and *K’* at the edge of the graphene Brillouin zone (BZ), called Dirac points, are essential for the physics of graphene. Their positions in momentum space can be expressed as
(3)$$K=\left(\frac{2\pi }{3a},\frac{2\pi }{3\sqrt{3}a}\right),\;K^{\prime }=\left(\frac{2\pi }{3a},-\frac{2\pi }{3\sqrt{3}a}\right).$$
The three nearest-neighbor vectors are expressed by
(4)$$\delta_{1}=\frac{a}{2}\left(1,\sqrt{3}\right),\delta_{2}=\frac{a}{2}\left(1,-\sqrt{3}\right),\delta_{3}=-a\left(1,0\right)$$
while the six second-nearest neighbors are positioned at $\delta \prime_{1}=\pm a_{1},\;\delta \prime_{2}=\pm a_{2},\;\mathrm{and}\;\delta \prime_{3}=\pm (a_{2}-a_{1})$.

The graphene electronic structure can be predicted by graphene simulation, especially using density functional theory (DFT). The adsorption energy of graphene and other materials (such as lithium, sodium, hydrogen and potassium), density of states (DOS), geometry, work function and dipole moment of each adatom-graphene system are studied and calculated using DFT [28,29,30,31,32,33,34]. 

#### 2.1.2. Ambipolar Electric Field Effect in Single-Layer Graphene

As shown in Figure 1b, the exceptional quality of graphene can be precisely seen in its pronounced ambipolar electric field effect [14]. Charge carriers can have high mobilities μ over 15,000 cm^2^ V^−1^ s^−1^ at ambient conditions and can be continually switched between electrons and holes in concentrations n of 10^13^ cm^−2^ [4,5,6,7]. The mobility of graphene has a consistently high value despite the high n (10^13^ cm^−2^) in electrically and chemically doped devices.

#### 2.1.3. Quantum Hall Effect

The QHE is another factor indicating the system’s outstanding electronic quality. Because the temperature range of the QHE for graphene is 10 times broader than that of other 2D materials, the QHE in graphene can be seen at room temperature. The unique reaction of massless fermions in a magnetic field is made more evident by their behavior at the high-field limit, where Shubnikov-de Haas oscillations (SdHOs) progress to the QHE [7]. In Figure 1c, the Hall conductivity $\sigma_{xy}$ and longitudinal resistivity $\rho_{xx}$ of graphene are shown with electron or hole concentrations in a constant magnetic field B. Pronounced QHE plateaus are detectable but do not behave in the expected progression $\sigma_{xy}=\left(4e^{2}/h\right)N$, where *N* is an integer. The first plateau occurs at $\left(2e^{2}/h\right)$, and the progression is $\left(4e^{2}/h\right)\left(N+1/2\right)$ because the plateaus correspond to half-integer ν values. In graphene, the movement from the lowest hole (ν = -1/2) to the lowest electron (ν = +1/2) Landau level (LL) needs the same amount of carriers ($\Delta n=4B/\varphi_{0}\approx 1.2\times 10^{12}{\mathrm{cm}}^{-2}$), as does movement between other adjacent levels. To highlight this extraordinary property, Figure 1c shows the $\sigma_{xy}$ of a graphite film comprising only two graphene layers. The existence of a quantized level at zero E, which is shared by electrons and holes, is the most important factor in realizing the unusual QHE sequence [14,35,36,37,38,39] (Figure 1d(1)). Another means of describing the half-integer QHE is to identify the coupling between pseudospin and orbital motion [5,7,40]. The pseudospin related to massive Dirac fermions induces a geometrical phase of 2π, which appears in the double degeneracy of the zero-E LL (Figure 1d(2)). Interestingly, the ‘standard’ QHE with all the plateaus present can be recovered in bilayer graphene by the electric field effect. A gate voltage causes an asymmetry between the two graphene layers, resulting in a semiconducting gap. The electric-field-induced gap results in a sustained QHE sequence by dividing the double step into two (Figure 1d(3)).

#### 2.1.4. Visual Transparency of Graphene

The transparency or opacity of suspended graphene is dependent only on the fine-structure constant, $\alpha =e^{2}/hc\approx 7.299\times 10^{-3}$ (where *c* is the speed of light) [14]. This parameter, conventionally related to quantum electrodynamics, defines the binding of light and relativistic electrons. Nair et al. reported that graphene absorbs a large portion (πα=2.3%) of incident white light despite being only one atom thick [20]. The universal G (high-frequency conductivity) for Dirac fermions means that detectable quantities such as graphene’s optical transmittance *T* and reflectance *R* are also universal (expressed by T≡(1+2πGc)−2=(1+12πα)−2 and R≡14π2α2T, respectively, for normally incident light) [41,42]. Additionally, the opacity of graphene is yielded by (1−T)≈πα, which can be written by computing the absorption of light by two-dimensional Dirac fermions with Fermi’s golden rule. Figure 1e displays a graphene crystal sample covering submillimeter apertures in a metal scaffold (Figure 1e inset) in transmitted white light. The opacities of different areas can be compared because the schematic in Figure 1e shows only an aperture partially covered by suspended graphene. Differences in the observed light intensity can be explained by the line scan traveling across the image. Figure 1f illustrates an opacity of graphene of 2.3 ± 0.1% with minor reflectance (< 0.1%), in contrast to optical spectroscopy, which indicates that the opacity is nearly independent of the wavelength, λ. As shown in the inset of Figure 1f, the opacity increases by 2.3% for each graphene layer added to the membrane. The above results also demonstrate a universal dynamic conductivity $G=\left(1.01\pm 0.04\right)\;e^{2}/4\hbar$ (here *e* is the elementary charge and ℏ is the Planck constant) in the visible-frequency region, which is the expected performance for ideal Dirac fermions.

### 2.2. Electronic Applications

#### 2.2.1. Optical Devices

##### Photodetectors 

The ability of photodetectors to detect broad spectral ranges of light with high responsivity and fast photodetection is critical in optoelectronic applications ranging from sensing, imaging, and spectroscopy to communications. Therefore, numerous studies have reported the use of traditional semiconductors for broader ranges of photodetection, even in the mid-infrared to far-infrared regime. As a result, narrow-bandgap semiconductor compounds such as indium antimonide (InSb), mercury cadmium telluride (HgCdTe), lead sulfide (PbS), and lead selenide (PbSe) are now widely applied in mid-infrared or far-infrared photodetectors [43,44,45]. However, the cryogenic operation, high cost, and complicated growth processes of these narrow-bandgap semiconductors has limited their practical usage as broad-range photodetectors [46]. Therefore, graphene has advantages over narrow-bandgap semiconductor compounds in terms of its ultrabroad optical absorption spectrum and high room-temperature electron and hole mobilities. 

Liu et al. developed photodetectors consisting of a thin tunnel barrier sandwiched between two graphene layers [47]. The graphene/5-nm-thick Ta_2_O_5_/graphene heterostructure photodetector shows a remarkably high responsivity of ~1000 A W^−1^ under a low excitation power (~10^−9^ W) of 532 nm laser light at a 1 V source-drain bias voltage. Additionally, the authors fabricated a similar graphene/6-nm-thick intrinsic silicon/graphene heterostructure device with improved photoresponsivity in the infrared regime. The intrinsic-silicon-incorporated photodetector shows high responsivity in a broadband regime ranging from near-infrared (4 A W^−1^ at 1.3 μm light and 1.9 A W^−1^ at 2.1 μm light) to mid-infrared light (1.1 A W^−1^ at 3.2 μm). Yu et al. fabricated a mid-infrared photodetector by hybridizing graphene with an innovative narrow-bandgap semiconductor, titanium sesquioxide (Ti_2_O_3_) nanoparticles [48]. The schematic in Figure 2a shows the representative hybrid graphene/Ti_2_O_3_ nanoparticle photodetector, with a high responsivity of ~300 A W^−1^ and high detectivity of ~7 × 10^8^ cm Hz^1/2^ W^−1^ at room temperature. Furthermore, a high responsivity of ~120 A W^−1^ can be achieved from the photodetector over a wide range of wavelengths from 4.5 μm to 10 μm (Figure 2b). Figure 2c shows the superior performance of this material compared to those in other recent studies, such as hybrid ZnO/graphene [49], perovskite/graphene [50], QDs/graphene [51], etc. [52]. Wang et al. proposed seamless lateral graphene p-n junctions by performing selective-area ion implantation followed by the in situ chemical vapor deposition (CVD) of graphene [53]. The photodetector based on these seamless lateral graphene p-n junctions offers a high responsivity of 1.4~4.7 A W^−1^ and detectivity of ~10^12^ cm Hz^1/2^ W^−1^ in the broad spectral range of visible light (532 nm) to near-infrared light (1550 nm) under an illuminated light intensity of 15 mW/ cm^2^. Fang et al. proposed a graphene photodetector with a gold snowflake-like fractal metasurface design [54]. The authors measured the photovoltages generated when light was illuminated on the fractal metasurface and plain gold-graphene edge (*V*_fractal_ and *V*_edge_, respectively). The results showed that photovoltage enhancement factors (*V*_fractal_/*V*_edge_) of 8 to 13 over the visible-light range (476 to 647 nm) can be achieved. Cakmakyapan et al. fabricated a photodetector with gold-patched graphene nanostripes showing ultrafast photodetection and high responsivity [55]. The photodetector operates in a wide spectral range of visible light (800 nm) to infrared light (20 μm) with a high responsivity ranging from 0.6 A W^−1^ to 8 A W^−1^. Additionally, the photodetector shows an ultrafast photodetection speed exceeding 50 GHz. 

##### Light-Emitting Diodes

Solid-state light-emitting diodes (LEDs) are currently widely used in the semiconductor industry [56] due to their unique electronic properties. Recently, the improved performance of LEDs has enabled various electronic applications, such as smart displays, light communications, and optoelectronics [57,58,59,60]. The external quantum efficiency (EQE), current efficiency, and power efficiency are important factors that determine the performance of LEDs. Among these factors, the EQE is known to be dominated by the light extraction efficiency and internal quantum efficiency [61]. To date, the internal quantum efficiency of LEDs has been remarkably improved and has already reached the theoretical limit, which is larger than 90% [62]. In contrast, the light extraction efficiency has not reached the theoretical limit. Therefore, improving the light extraction efficiency is key to obtaining a higher EQE. A high optical transparency, high reliability and low sheet resistance are required for better light extraction efficiency. Thus, the use of transparent conductive electrodes (TCEs) is necessary; indium tin oxide (ITO) is one of the most widely used materials because of its high optical transmittance (above 80% at 450 to 550 nm) and low sheet resistance [63,64]. However, ITO has considerable disadvantages, which are its mechanical brittleness and high cost [65]. Thus, metal nanowires (NWs) and carbon-based materials such as carbon nanotubes and graphene have been investigated to replace ITO [22,66,67,68,69]. 

Zhang et al. fabricated four-inch flexible organic light-emitting diodes (OLEDs) showing a high brightness of ~10,000 cd m^−2^ by transferring ultraclean and damage-free graphene supported by rosin (C_19_H_29_COOH); Figure 2d shows the device structure [21]. To date, various materials, such as small organic molecules and macromolecular polymers, have been used as supporting layers for transferring graphene [73,74,75]. However, low solubility in all solvents and strong interactions with graphene make the elimination of the supporting layers after the transfer difficult. These problems damage and leave polymer residues on the transferred graphene layer, degrading the electrical and optical properties. Rosin shows high solubility in solvents and weak interactions with graphene, thus enabling an ultraclean and damage-free transfer. As a result, OLEDs with rosin-transferred graphene showed a maximum current efficiency of 89.7 cd A^−1^ and power efficiency of 102.6 lm W^−1^, exceeding the performance of ITO and poly(methyl) methacrylate (PMMA)-transferred graphene OLEDs (Figure 2e). In addition, the inset shows an optical image of a rosin-transferred graphene-based OLED showing uniformly bright green light. Lee et al. proposed flexible OLEDs with an electrode structure based on low-index hole injection layers (HILs) and high-index TiO_2_ layers sandwiching graphene electrodes [76]. This structure enables losses in both the surface plasmon polariton (SPP) and cavity resonance enhancement to be controlled, thereby leading to a high EQE. The proposed OLEDs showed ultrahigh EQEs of 40.8% and 62.1% for single- and multijunction structures, respectively. Additionally, the OLEDs were bendable to a radius of 2.3 mm because the TiO_2_ layer is capable of withstanding flexural strains up to 4%. Wang et al. developed a single tunable LED emitting light ranging from blue (~450 nm) to red (~750 nm) composed of semireduced graphene oxide sandwiched by graphene oxide (GO) and reduced graphene oxide (RGO) [77]. The LED produced red and green lights at 4.8 lm W^−1^ and 6000 cd m^−2^ under a 12 V bias voltage and a 0.1 A drive current and blue light at 0.67 lm W^−1^ and 800 cd m^−2^ under a 16.5 V bias voltage and a 0.1 A drive current. This experiment enabled in situ wavelength tuning by controlling the bias voltage and doping level. Huang et al. introduced transparent LEDs based on graphene-encapsulated Cu NWs [78]. Cu NWs are a strong candidate to replace ITO TCEs owing to their high transmittance (~93%), low sheet resistance (51 ohm/sq), and low cost [79,80]. However, the low stability against oxidation of Cu NWs limits their use as electrodes for LEDs. To solve this problem, gas-phase graphene can be used to encapsulate Cu NWs to enhance their stability: the material shows strong antioxidant stability while maintaining high optoelectronic properties (33 ohm/sq and 95% transmittance) over a broad transparency range (200~3000 nm). Therefore, the encapsulated NWs serve as electrodes by forming good ohmic contact with both n-GaN and p-GaN, thereby leading to the successful fabrication of blue-light LEDs. Seo et al. conducted similar research by integrating silver nanowires (AgNWs) with a graphene protecting layer to achieve high performance [81]. High-quality graphene was obtained by a two-step growth method, which results in reduced point defects and an increased graphene domain size. The measured sheet resistance of AgNWs with graphene produced via the two-step growth method (A-2GE) was 77.5 ± 10 ohm/sq, and that of AgNWs was 205.1 ± 40 ohm/sq. Additionally, the oxygen transmission rate (OTR) was measured after the graphene films had been transferred to a polyethylene terephthalate (PET) film. The OTR of 2-G/PET was 5.39 ± 0.9 cc/m^2^-day, 74% less than that of bare PET. Thus, A-2GE showed high optoelectronic properties and strong antioxidant stability and can be used for TCEs for LEDs. 

##### Solar Cells

Solar cells based on organic-inorganic perovskite materials have recently been widely studied due to their high power conversion efficiencies (PCEs), long-range carrier diffusion length, intense light absorption, and facile fabrication (low cost and low temperature) [82,83,84,85,86,87,88,89]. The active device layers of perovskite solar cells (PSCs) are sandwiched by electron and hole transport layers (ETLs and HTLs, respectively) [90,91,92]. Since the charge carrier transport properties can be affected by both the electronic structures of the various interfaces and the morphology of PSCs [93,94], graphene and related two-dimensional materials (GRMs), such as graphene quantum dots (GQDs) [95,96,97], RGO [97,98,99], and fullerene [100], have been widely used to tune the interfacial properties and morphologies of PSCs. GRMs have been exploited as the top electrodes of PSCs, as an interlayer between the ETL/HTL and the perovskite layer, and by integration with the ETL/HTL to provide efficient charge transfer [101,102]. 

You et al. proposed semitransparent PSCs with top graphene electrodes [103]. The fabricated PSCs showed maximum PCEs of 11.65 *±* 0.35% and 12.02 *±* 0.32% from the graphene and fluorine-doped tin oxide (FTO) sides, respectively, with optimized conditions of double-layer graphene and an ~70 mg mL^−1^ 2,2′,7,7′-tetrakis-(*N*,*N*-di-p-methoxyphenylamine)-9,9′-spirobifluorine (spiro-OMeTAD) solution due to the low surface roughness. Sung et al. first fabricated PSCs with over 17% efficiency by exploiting graphene as a transparent conducting anode [104]. The presence of molybdenum trioxide (MoO_3_) controlled the contact angle between poly(3,4-ethylenedioxythiophene):poly(styrene sulfonate) (PEDOT:PSS) and the graphene surface and helped successfully form PEDOT:PSS/MAPbI_3_ layers. The thickness of the MoO_3_ layer affected the PCE of the PSCs. In the experiment, a 1 nm-thick MoO_3_ layer with graphene-based devices showed a maximum PCEs of 17.1%, larger than 90% of those of ITO-based devices (18.8%). The flexibility and high PCE of this graphene-based device pave the way for the further development of flexible solar cells. Najafi et al. fabricated a PSC with a high PCE of over 20% using MoS_2_ QDs and a functionalized reduced graphene oxide (f-RGO) hybrid [70]. MoS_2_ QDs:f-RGO hybrids are used as both the HTL and active buffer layer (ABL), and a schematic of the PSC is shown in Figure 2f. The PSC structure consists of FTO/compact TiO_2_ (cTiO_2_)/mesoporous TiO_2_ (mTiO_2_)/MAPbI_3_/f-RGO:MoS_2_/spiro-OMeTAD/Au. The I-V characteristics of MoS_2_ QDs, f-RGO, and MoS_2_ QDs:f-RGO as ABLs are shown in Figure 2g. The PSC using MoS_2_ QDs:f-RGO showed a further improved performance, with a maximum PCE of 20.12% and a fill factor (FF) of 79.75%, compared to the reference device showing a maximum PCE of 16.85% and an FF of 76.9%. Agresti et al. fabricated a large-area PSC of 50.56 cm^2^ with a PCE of 12.6% by doping the mTiO_2_ layer with graphene flakes and inserting lithium-neutralized graphene oxide flakes (GO-Li) between the interface of the mTiO_2_ and perovskite [105]. The charge injection from perovskite to mTiO_2_ is improved by the GO-Li layer, and the graphene flakes dispersed in the mTiO_2_ layer help speed up the charge dynamic at the electrode. The interface engineering of PSCs by GRMs leads to an increased PCE, from 11.6% for the reference PSC to 12.6% for the PSC with GO-Li and mTiO_2_. Li et al. proposed a PSC with an MAI perovskite layer incorporating graphene nanofibers [106]. Integrating graphene nanofibers with perovskite leads to improved charge injection and the formation of larger crystals. Therefore, the PSCs with graphene nanofibers showed an improved PCE of 19.83% compared to the PCE of the reference device of 17.51%.

#### 2.2.2. Sensors

##### Electronic Sensors

Graphene can be utilized in various electronic sensors due to its unique device properties. Recently, novel integrated electronic circuitries with tactile pressure sensors based on piezoelectricity [107,108], piezoresistivity [109,110], capacitance [111,112], and field-effect transistors (FETs) [113,114,115] have been fabricated for healthcare and health-monitoring applications [107,108,115]. Haniff et al. successfully fabricated a flexible piezoresistive-type pressure sensor based on graphene synthesized at various substrate temperatures (750, 850 and 1000 °C) with hot-filament thermal chemical vapor deposition (HFTCVD) [116]. The authors found that the sensitivity of the flexible graphene-based piezoresistive-type pressure sensor could be tuned through this technique (graphene deposited at 750 °C has a four-fold higher sensitivity than graphene deposited at 1000 °C). A flexible and highly sensitive piezoresistive pressure sensor based on an ultrathin wrinkled graphene film (WGF), interdigital electrodes (IDEs), polyvinyl alcohol (PVA) NWs, and an interconnected isolation effect was reported by Liu et al. [71]. The sensing mechanism and current response to loading and unloading are described in Figure 2h. The WGF and interconnected PVA NWs produce a synergistic effect, resulting in a piezoresistive pressure sensor with an ultrasensitivity of 28.34 kPa^−1^ and mechanical durability and reliability (repeated tests of 6000 cycles). Shin et al. paved the way to use pressure sensors for diverse application areas, such as medical diagnosis, robotics and automatic electronics [117]. The authors developed transparent tactile pressure sensors covering a wide pressure range (250 Pa~3 MPa) by forming fully integrated active-matrix pressure-sensitive graphene FET arrays with air-dielectric layers between two folded opposing panels.

For the true integration of woven electronics and/or optoelectronics into textiles, the direct fabrication of the device on the fiber itself with high-performance materials allowing easy unification into fabrics is required [118,119]. Alonso et al. completely integrated flexible (up to 10 mm, the radius of a human finger), transparent and durable graphene-based LED and touch sensors on textile fibers [120]. Roll-to-roll and printing-compatible patterning techniques were adopted to fabricate the LED and the capacitive touch sensors. Transparent electrodes coated by monolayer graphene on a textile fiber can be utilized for wearable electronics with high conductivity and flexibility [121]. Graphene-based optical waveguide tactile sensors can overcome the disadvantage of optical sensors with a directional coupler [122]. Additionally, transparent and flexible UV sensors play an important role in the field of wearable and/or portable optoelectronic systems. Pyo et al. demonstrated a fully transparent, exceedingly sensitive, and flexible UV sensor with 1D hybrid carbon nanotubes (CNTs) combined with a 2D graphene electrode [123]. Because of the provided effective charge transfer and the minimized effect of contact resistance by the integration of the CNTs and graphene without a potential barrier, the CNT–graphene UV sensors have a 30 times higher photoresponse (45% under 254 nm UV illumination with a power intensity of 1.91 mW cm^−2^) than that of the CNT-Au electrode sensors. Additionally, thanks to the outstanding properties of graphene and CNT, the device shows a high optical transparency (over 80% at 550 nm) and remarkable mechanical flexibility (bending radius of 5.5 mm) with high electrical reliability. Goossens et al. reported the monolithic integration of graphene with a complementary metal–oxide–semiconductor (CMOS)-integrated circuit operating as a high-mobility phototransistor [124]. Broadband high-resolution image sensing and a sensitive digital camera in the range of ultraviolet, visible and infrared light was demonstrated.

Research on the safe and accurate monitoring of human body signals, hazardous disease detection, and stimulation systems has been highly focused in areas ranging from biological research to clinical medicine [125,126,127,128,129,130,131,132]. Electrooculography (EOG) is a technique for recording the corneo-retinal standing potential between the retina and the cornea of human eyes. Ameri et al. described a noninvasive graphene electronic tattoo (GET)-based invisible EOG sensor [133]. Ultrathin (350 nm thickness), stretchable (up to 50%), soft, fully transparent (85% in the visible regime), and breathable EOG sensors based on GETs have a high resolution (approximately 4° to detect eye movement); wireless communication can be achieved by connecting the sensor to an OpenBCI Cyton board, and the sensors can be successfully applied to a human–robot interface (HRI). Kuzum et al. simultaneously recorded electrophysiological spikes in the brain with flexible, fully transparent graphene-based neural electrodes and neuro-optical images with confocal or multiphoton microscopy [134]. The graphene electrodes were highly transparent, such that both electrophysiological recording and optical imaging, leading to brain imaging with high spatiotemporal dynamics, could be achieved without perturbing either technique. Additionally, the simultaneous optical stimulation of neural tissue and electrical recording of the brain can be achieved by using µ-LEDs and transparent graphene electrodes, respectively. [135]. Such powerful graphene-based systems that resolve individual cells and their connections through optical imaging and electrical recording of brain activity are a breakthrough in neuroscience. Monitoring glucose [136,137] and the intraocular pressure [138,139,140,141] are particularly crucial for detecting and managing diabetes and glaucoma, respectively. Kim et al. fabricated a highly conductive, flexible and transparent wearable smart contact lens using a reliable and robust hybrid structure of 1D and 2D nanomaterials [72]. Wearable contact lenses can directly and noninvasively detect and wirelessly monitor biomarkers such as the glucose contained in tears and intraocular pressure by measuring the change in resistance and capacitance of the electronic device. Figure 2i shows a schematic of fully integrated multifunctional sensors on a soft contact lens. As shown in Figure 2j, a lower optical transmittance, haziness, and sheet resistance were obtained when graphene and AgNWs were integrated in a hybrid system than when the single materials of graphene or AgNWs were used.

##### Biomolecule Sensors

For early and reliable clinical cancer diagnosis and treatments, cost-efficient, dependable and sensitive monitoring and detecting systems are crucial. Wu et al. fabricated a functionalized graphene-based electrochemical sensor array system for cell sensing [142]. The graphene-based chemical nose/tongue approach system can discern 100 cell samples consisting of (i) artificial circulating tumor cells (CTCs); (ii) cancerous, multidrug-resistant cancerous and metastatic human breast cells; and (iii) different cell types with almost 100% classification accuracy.

Detecting and monitoring various electrolytes, such as biological, organic, and inorganic electrolytes, and electroactive materials in human body fluids are important for clinical diagnosis and analytical applications [61,143,144,145]. Wang et al. developed electrochemical biosensors to selectively and simultaneously detect five analytes in human serum: ascorbic acid (AA), dopamine (DA), nitrite (NO_2_), tryptophan (Trp), and uric acid (UA) [146]. A CVD method was used to fabricate freestanding graphene nanosheets on tantalum (Ta) wire. Graphene-based biosensors provided highly selective, sensitive results (detection limits of AA, DA, NO_2_, Trp, and UA of 1.58, 0.06, 6.45, 0.10, and 0.09 μM, respectively (S/N = 3)) with differential pulse voltammetry (DPV). As predicted by the World Health Organization (WHO), the number of diabetes patients has been gradually increasing [147], and such patients require the continuous monitoring of glucose levels in the interstitial fluid. Currently, only invasive methods are available for monitoring the glucose in the blood [148,149,150]. Lipani et al. demonstrated a noninvasive, transdermal, path-selective, and specific glucose monitoring and detection system based on a miniaturized device array platform fabricated by a graphene thin-film method or screen-printing technology [151]. The authors found via in vivo testing on healthy humans that the graphene-based glucose monitoring system can continuously record the blood sugar for 6 h. 

The most important consideration for systems biology and personalized and precision medicine is an acceptable affinity and binding efficiency of deoxyribonucleic acid (DNA) hybridization and the ability to distinguish DNA sequences with single-nucleotide substitutions. Xu et al. suggested next-generation multichannel graphene-based DNA sensors for a cost-effective, fast, simple and label-free biosensing system to perform kinetic studies (e.g., detection limit of 10 pM for DNA) [152]. Reproducible and reliable G-FET DNA biosensors have been designed using a mature FET fabrication method characterized by low cost, low power, and simplicity of miniaturization. Diffusive transport is a general mechanism for biomolecules in viscous media, and a crucial prerequisite for biosensing is placing the target biomolecules at the most sensitive point. Because of the powerful dielectrophoresis (DEP) forces of monolayer graphene, the sharp edge of monolayer graphene can successfully produce singular electrical field gradients, accurately trapping and positioning biomolecules (nanoparticles and DNA molecules) [153]. Seo et al. fabricated a sensitive and selective electrochemical genosensor integrated with grown graphene films, realizing reliable biodetection and demonstrating the functionality of the graphene films [154]. 

A provisional WHO guideline has established a concentration limit of microcystin-LR (MC-LR) of 1 μg/L in drinking water [155] because the continuous monitoring of drinking water quality and appropriate treatment are extremely important for human healthcare and the quality of life worldwide [156,157]. Zhang et al. reported a novel graphene film-integrated biosensor for MC-LR detection that is time- and cost-effective, portable, scalable to large-scale manufacturing, and easy to handle. A graphene film/polyethylene terephthalate (GF/PET) composite grown through the CVD method was used to develop a graphene-based biosensor, resulting in a detection limit of 2.3 ng/L, which fully satisfies the safety guideline of the WHO. 

##### Gas Molecule Sensors

With fast-growing industrial development, toxic gases endangering human beings have been extensively produced and emitted [158,159]. Therefore, developing gas sensors with a high detection sensitivity or control capacity for hazardous gases is extremely important for the protection of human health and the environment [160]. Wu et al. reported a chemiresistor-type sensor based on a 3D sulfonated reduced graphene oxide hydrogel (S-RGOH); the sensor is capable of detecting a variety of important gases with high sensitivity, boosted selectivity, fast response, and good reversibility [161]. The NaHSO_3_-functionalized RGOH displays 118.6 and 58.9 times higher responses than its unmodified RGOH counterpart for NO_3_ and NH_3_, respectively. Moreover, the response increases monotonically from 6.1% at 200 ppb NO_2_ to 22.5% at 2 ppm NO_2_. Guo et al. fabricated a NO_2_ gas sensor from the room-temperature reduction of GO via two-beam-laser interference (TBLI) [162]. The fabricated RGO sensor enhanced the sensing response for NO_2_ and accelerated the response/recovery rates. For 20 ppm NO_2_, the response (*R*_a_/*R*_g_) of the sensor based on RGO hierarchical nanostructures is 1.27, which is higher than those of GO (1.06) and thermally reduced RGO (1.04). The response time and recovery time of the sensor based on laser-reduced RGO are 10 s and 7 s, respectively, which are much shorter than those of GO (34 s and 45 s). 

Because of the significant properties (high heat of combustion and low minimum ignition energy) of hydrogen (H_2_), it can be exploited in various applications [163,164]. However, because of its delicate properties (low level of explosion limit ~4%), the real-time and long-distance monitoring of H_2_ concentrations is essential. Sharma et al. produced an FET hydrogen sensor integrated with a graphene-Pd-Ag-gate FET (GPA-FET) [165]. The GPA-FET showed an excellent sensing response to hydrogen gas at 25~254.5 °C.

Many studies have reported the integration of graphene-based mobile gas sensors with modern technologies such as smart phones, cloud computing and the Internet of Things (IoT) with graphene nanoribbons due to the high electrical conductivity of the nanoribbons. The first monolithically integrated CMOS-monolayer graphene gas sensor was developed by Zanjani et al., with a minimal number of post-CMOS processing steps, to realize a gas sensor platform that integrates the higher gas sensitivity of monolayer graphene with the low power consumption and cost advantages of a silicon CMOS platform [166]. Furthermore, Pour et al. improved graphene nanoribbons’ electrical conductivity by lateral extension [167]. Improved graphene nanoribbons were synthesized in solution; the lateral extension decreased their bandgap and improved their electrical conductivity. 

## 3. Thermal Properties and Applications

### 3.1. Thermal Properties 

#### 3.1.1. Specific Heat of Graphene and Graphite 

The specific heat, *C*, is a distinct characteristic of a material expressing the change in the energy density *U* with respect to the change in temperature (1 K or 1 °C), represented as *C* = d*U*/d*T*, with units of joules per kelvin per unit mass, per unit volume, or per mole. The thermal time constant of a material, which indicates how quickly the material heats or cools, is determined by the specific heat. The thermal time constant is expressed as τ ≈ *RCV*, where *V* is the volume of the material and *R* is the thermal resistance for heat dissipation [168]. As the specific heat of graphene has not been measured directly, the calculation was performed by referencing the data for graphite [169,170,171]. The specific heat of a material is contributed by two components, phonons (or lattice vibrations) and electrons: *C* = *C*_ph_ + *C*_el_. However, the contribution of phonons dominates the specific heat of graphene at all temperatures [172]. In addition, as shown in Figure 3a, the phonon specific heat increases with the temperature [171,173]. The specific heat of graphite at room temperature is *C*_ph_ ≈ 0.7 J g^−1^ K^−1^, which is approximately 30% higher than that of diamond due to the higher density of states (DOS) at low phonon frequencies caused by the weak coupling of graphite layers. For a graphene sheet at room temperature, a similar result is expected, but the specific heat can be altered when graphene interfaces with a substrate (e.g., graphene on insulators) [174]. 

#### 3.1.2. Thermal Conductivity

The thermal conductivity *K* is explained by Fourier’s law, q=−K∇T [175]. In this equation, the negative sign means heat flows from high to low temperature; *q* is the heat flux per unit area; and ∇T is the temperature gradient. The thermal conductivity is related to the specific heat by $K\approx \sum^{\text{​}}C\nu \lambda$, where *ν* is the average phonon group velocity and *λ* is the mean free path. Therefore, as the specific heat of graphene is dominated by phonon transport, the thermal conductivity is also dominated by phonon transport [171].

Here, the thermal conductivity of various carbon allotropes, including two types of pyrolytic graphite (in-plane and cross-plane), diamond, and amorphous carbon, is presented in Figure 3b. Pyrolytic graphite is similar to highly oriented pyrolytic graphite (HOPG). The in-plane *K* of pyrolytic graphite is approximately 2000 W m^−1^ K^−1^ at room temperature, and the cross-plane *K* at room temperature is more than two orders of magnitude smaller. Because HOPG is composed of well-aligned large crystallites, the overall behavior is analogous to that of a single crystal; this fact explains the difference in *K* [177]. Heat is mainly transported by acoustic phonons in all bulk carbon allotropes. In HOPG and diamond, *K* reaches maximum values at approximately 100 K and 70 K, respectively. In amorphous carbon, *K* ranges from 0.01 W m^−1^ K^−1^ to 2 W m^−1^ K^−1^ at 4 K and 500 K, respectively. Similarly, carbon allotropes have different thermal conductivities depending on the temperature. The difference is attributed to the grain size and quality of graphite and the phonon DOS, as indicated by *C*_ph_. 

A method to measure the thermal conductivity of graphene by exploiting confocal micro-Raman spectroscopy is introduced in Figure 3c. The G peak in the graphene spectra depends strongly on the temperature [178]. This high temperature sensitivity of the G peak enables the monitoring of the local temperature change induced by laser light focused on a graphene layer. In this experiment, trenches were fabricated on Si/SiO_2_ substrates by reactive ion etching (RIE), and graphene was suspended over the trenches. The depth of the trenches was 300 nm, and the width varied from 2 to 5 μm [176]. An optical image of the trenches taken by scanning electron microscopy (SEM) is shown in Figure 3d. The laser light focused on the middle of the suspended graphene layer generates heat in the graphene. The heat generated by laser excitation propagates laterally through the graphene due to the negligible thermal conductivity of air. Thus, even a small amount of heat propagated from the middle of the graphene can result in a detectable temperature increase. The heat front propagating through the graphene layers can be explained by two components: the plane-wave heat front and the radial heat wave [179]. The thermal conductivity from the plane-wave heat front can be expressed as K = (L/2S)(ΔP/ΔT), where ΔP/ΔT indicates the heating power change with respect to the temperature change, L is the distance from the center of the graphene to the heat sink, and S = h × W. For the radial heat wave case, $K=\chi_{G}\left(1/2h\pi \right){\left(\delta \omega /\delta P\right)}^{-1}$, where δω/δP indicates the G peak position shift due to the heating power change. Consequently, the thermal conductivity can be expressed as
(5)$$K=\chi_{G}\left(L/2hW\right){\left(\delta \omega /\delta P\right)}^{-1}.$$

As shown in Figure 3e, the excitation power dependence of the Raman G peak was measured for the suspended graphene layers. The increase in laser power induced an increase in the intensity and redshift of the G peak. The G peak position shift with respect to the power change in the suspended graphene layers is shown in Figure 3f. The slope from the measured data is δω/δP_D_ ≈ −1.29 cm^−1^mW^−1^, where P_D_ is the total dissipated power and the temperature coefficient $\chi_{G}\;=\;-1.6\;\times \;10^{-2}\;{\mathrm{cm}}^{-1}/K$ [180]. When these values are substituted into equation (5), a thermal conductivity of approximately 2000~4000 W m^−1^ K^−1^ is obtained for freely suspended graphenes; this value is the highest of any known material [15,16,17]. Raman spectroscopy is a basic technique for graphene analysis. In addition to thermal conductivity, Raman spectroscopy can help to characterize graphene flakes [181], detect traces of molecules [182] and achieve electronic properties [183]. 

#### 3.1.3. Thermoelectric Effects of Graphene

The thermoelectric power (TEP) is the voltage induced by a temperature gradient. Experimental studies indicated that graphene has a TEP of ~ 50 to 100 μV K^−1^ [184]. In addition, other experiments showed that graphene has a maximum TEP value of ~80 μV K^−1^ at room temperature. It was theoretically verified that TEP behaves as $1/\sqrt{n_{0}}$ at a high carrier density (*n*_0_) but is saturated at low densities. The values of the Seebeck coefficient range from ~100 to ~10 μV K^−1^ for temperatures ranging from ~100 to ~300 K. The Seebeck coefficient (S=dV/dT) shows lower values at high temperatures, as shown in Figure 3g [18]. Graphene has a higher *S* than those of semiconductors, and the polarity of *S* can be controlled by varying the gate voltage; these thermoelectric effects are intriguing [185]. The figure of merit (*ZT*) of the efficiency of thermoelectric energy conversion is defined as $ZT=S^{2}\sigma T/\left(K_{\mathrm{ph}}+K_{el}\right)$, where σ is the electrical conductivity [186]. Graphene shows a high value of *ZT* due to suppressed phonon scattering, which leads to decreased *K* ($K_{\mathrm{ph}}+K_{el}$) [187,188]. The high value of *ZT* for graphene suggests the possibility of its use in energy-harvesting applications. The *ZT* of various graphene nanostructures is very promising for the improvement of the thermoelectric energy conversion [189]. 

### 3.2. Thermal Applications

#### 3.2.1. Sensors

##### Temperature Sensors

Electronic skins (E-skins) can generate electronic signals from external stimulation for use as human-machine interfaces (HMIs) and multifunctional smart skins [113,190,191,192,193,194]. In addition to pressure and strain sensing, achieving the simultaneous detection and monitoring of temperature with cost-efficient fabrication is important to make plausible mimics of multifunctional human skin. Ho et al. fabricated a fully transparent (over 90% transmittance in the range 400 ~ 1000 nm) and stretchable graphene-based multifunctional E-skin sensor matrix [195]. The matrix can detect and monitor humidity, temperature, and pressure through a simple lamination process. Figure 4a shows the fabrication technique of the sensor matrix. CVD-grown graphene interconnects electrodes with three sensors, and GO and rGO are the active sensing materials for humidity and temperature, respectively. Figure 4b shows the resistance change with respect to temperature, and real-time measurement results are shown. Each sensor was simultaneously and individually sensitive to only its relevant form of stimulation, and no response was obtained from other forms of stimulation. Vuorinen et al. reported a simple technique to fabricate graphene/PEDOT:PSS-based skin-conformable inkjet-printed temperature sensors [196]. A Phene Plus I3015 transparent graphene/PEDOT:PSS ink was printed onto polyurethane skin-conformable adhesive bandages to form transparent and flexible temperature sensors. The inkjet-printing method can significantly reduce the manufacturing costs and wasted materials and provides advantages for the production of disposable systems. The graphene/PEDOT:PSS temperature sensors present remarkable sensitivity to monitor temperature changes on the human skin with a temperature coefficient of resistance (TRC) higher than 0.06% per degree Celsius. Additionally, Trung et al. demonstrated an all-elastomeric transparent stretchable (TS) gated sensor with a simple method using graphene [197]. The schematic of the structure of the TS-gated sensor is shown in Figure 4c. The overall system integrated a graphene-based temperature sensor and strain sensor. A PEDOT:PSS/PU dispersion (PUD) composite elastomeric conductor serves as the source, drain, and gate contacts. A PU R-GO/PU nanocomposite and a composite of AgNWs/PEDOT:PSS/PUD as a gate dielectric are used to form a temperature-responsive channel layer and a strain sensing layer, respectively. The sensor shows a high stretchability of up to 70%, responsivity of up to 1.34% per 34 °C and durability (10,000 test cycles at a strain of 30%) and could be conformally attached to human body skin to monitor temperature changes. Figure 4d indicates the sensor’s ability to monitor the surface temperature of targets: cold and hot water. The dashed line illustrates the device region. Such devices can be useful for applications including interactive remote healthcare systems, biomedical monitoring, and HMIs by integrating wireless communication units such as Bluetooth into the sensors.

Single-layer graphene (SLG) presents impressive thermal properties, so graphene can be used for thermal management and temperature sensing applications. Davaji et al. fabricated and compared micropatterned SLGs on a silicon dioxide (SiO_2_)/Si substrate, a silicon nitride (SiNx) membrane, and a Si wafer with etched rectangular boxes (10 × 20 μm^2^) [198]. These sensors show a quadratic dependence of the resistance versus temperature in the range of 283 ~ 303 K, as determined through analyzing the temperature-dependent carrier density, electron mobility relationship (~T^−4^) and electron-phonon scattering. The graphene-based sensor fabricated on a SiNx membrane has a substantially faster response, better sensitivity because of the low thermal mass, and better mechanical stability than other suspended graphene sensors. As a result, a temperature sensor comprising the suggested SLG on a SiNx membrane can be exploited for various highly sensitive and fast applications. 

##### Thermoelectric Sensors

Graphene can be utilized as not only a temperature sensor but also a sensor of other crucial parameters. Though terahertz radiation has been used in various applications ranging from security to medicine [201], the sensitive room-temperature detection of terahertz radiation is very difficult [202]. Cai et al. successfully demonstrated a graphene thermoelectric terahertz photodetector with highly impressive sensing ability (over 10|V|W^−1^ (700|V|W^−1^) at room temperature) and low noise-equivalent power (below 1100|pW|Hz^−1/2^ (20|pW|Hz^−1/2^)) [203]. The compared reference was the incident (absorbed) power. The detection mechanism of the sensor is the hot-electron photothermoelectric effect in graphene. Because of strong electron-electron interactions, photoexcited carriers quickly gain heat [204,205], and the energy of the lattice decreases more slowly [204,206]. Electron diffusion is caused by the temperature gradient of electrons. As a results, a net current is generated by asymmetry due to local gating [207,208] or dissimilar contact metals [209] through a thermoelectric effect. The operation of graphene-based sensors is similar to that of state-of-the-art room-temperature terahertz detectors [210], and the ability of the former to obtain time-resolved measurements is eight to nine times faster than that of the latter [208,211]. Additionally, one of the important considerations in on-chip nano-optical processing is the detection, control and generation of propagating plasmons [212,213,214]. Graphene shows intensely confined and controllable (>0.5 ps) plasmons induced by electrostatic fields [215,216,217,218]. An all-graphene-based mid-infrared plasmon detector that precisely converts the natural decay product of the plasmon (electronic heat) to voltage via the thermoelectric effect [219,220] was presented by Lunderberg et al. [221]. The detection system, constructed with the plasmonic medium of graphene encapsulated in hexagonal boron nitride (hBN), operates at room temperature; a single graphene sheet simultaneously acts as the plasmonic medium and detector. The junction induced by two local gates was used to completely control the thermoelectric and plasmonic behavior of the graphene.

##### Thermal Biosensors

Recently, studies aiming to realize theranostic systems with simultaneous detection and therapeutic services in one single nanostructure have increased in the field of personalized nanomedicine and clinical biomedical applications [222,223,224]. Cao et al. reported a multifunctional theranostic system coupling diagnostic and therapeutic functions with a porphyrin derivative (P) and GQDs [199]. The P is capable of high singlet oxygen production, and GQDs have excellent fluorescence properties. A GQD- polyethylene glycol (PEG)-P system was fabricated by integrating the P into PEGylated and aptamer-functionalized GQDs; the system showed exceptional physiological stability, impressive biocompatibility and low cytotoxicity. The intrinsic fluorescence of the GQDs can distinguish cancer cells from somatic cells and helps to detect intracellular cancer-related microRNA (miRNA). The photothermal conversion properties of GQD-PEG-P are shown at the bottom of Figure 4e. The image results indicated that the temperature of 100 μg/mL GQD-PEG-P solution increased to 53.6 °C, while the temperature of control water remained at 33.2 °C. A cancer treatment system combining progressive photothermal therapy (PTT) and powerful photodynamic therapy (PDT) demonstrated outstanding efficiency for both in vitro cancer cells and in vivo multicellular tumor spheroids (MCTS). A total GQD-PEG-P theranostic system for intracellular miRNA detection and PTT and PDT therapy is presented at the top of Figure 4e (the molecular beacon (MB) is the detection probe for miRNA-155). After monitoring and detecting a cancer, highly effective and reliable treatment is crucial. Synergistic cancer cures such as drug delivery, magnetic hyperthermia, photothermal therapy, gene therapy, and radiotherapy have been spotlighted [225,226,227,228,229,230]. To accomplish safe and successful treatment, Yao et al. suggested a multifunctional platform with GQD-capped magnetic mesoporous silica nanoparticles (MMSNs); a composite of GQDs was used for caps and local photothermal generators, and MMSNs were used for controlled drug release and magnetic hyperthermia [231]. Drug release can be handled by changing the interaction between drug molecules and carriers and by controlling the state (open or closed) of the outlets of mesoporous channels [232,233]. Monodisperse MMSN/GQD nanoparticles (particle size of 100 nm) can carry doxorubicin (DOX), provoke DOX discharge and adequately generate heat in an alternating magnetic field (AMF) or by near-infrared irradiation. An integrated graphene-based chemo-magnetic hyperthermia therapy or chemo-photothermal therapy platform for destroying breast cancer 4T1 cells (the model cellular system) represents a remarkable synergistic effect. As a result, the MMSN/GQD nanoparticles have excellent potential for efficient and accurate cancer therapy.

#### 3.2.2. Energy Management Systems

Efficient heat management systems have become extremely important in various fields, such as electronic, optoelectronic, and thermoelectric applications, with the fast-growing development of state-of-the-art technologies to transport heat to the field environment; a high power heating density is required, or excessive heat must be prevented from deteriorating the device reliability, lifetime, and performance [234,235]. In this regard, the development of materials with high thermal conductivity is urgently needed. Heat management systems are normally based on metallic materials such as Al and Cu because of their relatively high thermal conductivity, approximately 200 to 400 W m^−1^ K^−1^, and low cost [236,237]. However, these materials show insufficient thermal conductivity when fabricated with nm thickness due to the linear relation between the thickness and thermal conductivity [238]. Therefore, graphene, which has superior thermal conductivity, is a promising alternative for use in practical heat management systems. Seo et al. studied the pool boiling of hybrid graphene/single-walled carbon nanotubes (SWCNTs), graphene, and SWCNT films deposited on ITO surfaces to prove that the hybrid graphene/SWCNT material is the layer with the most enhanced heat transfer coefficient [200]. The authors tested four types of heating surfaces: (1) a bare ITO surface, (2) SWCNTs, (3) graphene, and (4) hybrid graphene/SWCNT layers deposited on an ITO surface. CVD was used to deposit graphene on the ITO surface, and SWCNTs were spray-coated on the surface to fabricate hybrid graphene/SWCNTs. The critical heat flux (CHF) of the SWCNTs, graphene, and hybrid graphene/SWCNT heaters were measured to be 123.0, 130.5, and 141.6 kW/m^2^, respectively. The CHF of the hybrid graphene/SWCNT heater showed the greatest improvement of 18.2% compared to that of the bare ITO heater. As shown in Figure 4f, the maximum heat transfer coefficient (HTC) values of the bare ITO, SWCNT, graphene, and hybrid graphene/SWCNT heaters were found to be 4.41, 5.31, 4.49, and 6.83 kW/m^2^ K, respectively. The HTC of the hybrid graphene/SWCNT heater improved by 55% compared to that of the bare ITO heater. These improvements are attributed to the deposition of SWCNTs on the graphene, which compensates the disconnected areas and wrinkles of the graphene. Han et al. introduced a high-performance heat spreader to manage excessive heat [239]. Graphene-based films (GBFs) were used in this experiment because of their high thermal conductivity of 1600 Wm^−1^ K^−1^ [240]. With a heat flux of 1300 W cm^−2^, the temperature of the hotspot decreased by 17 °C with a GBP deposited on nonfunctionalized GO. In addition, 3-amino-propyltriethoxysilane (APTES) functionalization resulted in an additional decrease in temperature of 11 °C, or a total decrease of 28 °C. Bernal et al. proposed edge-functionalized graphene nanoplatelets (GnPs) to improve the thermal conductivity [241]. The diazonium reaction was exploited to synthesize phenol-functionalized GnPs (GnP-OH) and dianiline-bridged GnPs (E-GnPs). The in-plane thermal conductivity of both GnP-OH and E-GnPs was enhanced by 20% compared to that of GnPs. Even the cross-plane thermal conductivity of E-GnPs was remarkably increased by 190% compared to that of GnPs. Ma et al. conducted research on the relation between the thermal conductivity and grain size of graphene [242]. Segregation adsorption CVD (SACVD) was exploited to grow high-quality graphene films with a controllable grain size ranging from ~200 nm to ~1 μm. The experimental results demonstrated that the thermal conductivity increased exponentially from ~610 to ~5230 W m^−1^ K^−1^, with grain sizes ranging from ~200 nm to 10 μm. The thermal conductivity of graphene was found to be significantly decreased with the decreasing grain size by a small thermal boundary conductance of ~3.8 × 10^9^ W m^−2^ K^−1^. Ding et al. proposed graphene nanosheets (GNSs) exhibiting high dispersibility in water and thermal conductivity [243]. Graphite powder and a modifier of sodium lignosulfonate (LS) were used to fabricate the GNSs by a ball-milling method. The GNSs fabricated with this method exhibited a high thermal conductivity of 1324 W m^−1^ K^−1^. 

Currently, with the rapid development of industry, the energy consumption of mainly fossil fuels is causing dramatic depletion worldwide [244]. Additionally, various environmental problems, such as desertification, the greenhouse effect, and pollution, are accompanied by the excessive use of energy [245,246]. In this regard, the development of sustainable energy is urgently required, and heat, which is approximately 60% of all energy waste, is rising as a promising candidate [247]. Thus, thermoelectric materials are of high interest. As mentioned above, the figure of merit ZT is defined as ZT = S^2^σT/(K_ph_ + K_el_); thus, enhancing the Seebeck coefficient and electrical conductivity and reducing the thermal conductivity are essential to achieving a higher efficiency. Cho et al. proposed multidimensional nanomaterials based on organic thin films exhibiting a high thermoelectric power factor at room temperature [248]. The thin film consists of the organic materials polyaniline (PANi), PEDOT:PSS-stabilized graphene, and PEDOT:PSS-stabilized double-walled nanotubes (DWNTs). These materials are deposited as a quad-layer of PANi/graphene-PEDOT:PSS/PANi/DWNT. The quad-layer film showed a gradual increase in the electrical conductivity and Seebeck coefficient with the increased number of layers. The electrical conductivity and Seebeck coefficient of the film were found to be 1.9 × 10^5^ S m^−1^ and 120 μV K^−1^ at 80 QLs, respectively. These results led to a high value of the thermoelectric power factor (PF) (~2710 μW m^−1^ K^−2^), where PF = S^2^σ. Ma et al. doped bromine (Br) onto graphene fibers to enhance the thermoelectric properties [27]. The porous structure of the Br-doped graphene fibers enhances phonon scattering and leads to limited thermal conductivity compared to that of other graphene materials [249,250]. Additionally, doping with Br induces a downshift in the Fermi level, which causes an increased Seebeck coefficient and increased electrical conductivity [251,252]. As a result, the experiment exhibited a maximum figure of merit of 2.76 × 10^−3^ and room-temperature PF of 624 μW m^−1^ K^−2^. These values are better than those of materials composed of solely graphene and CNTs. Lin et al. observed the enhanced thermoelectric effect of lanthanum strontium titanium oxide (LSTO) upon adding graphene [253]. The addition of graphene to LSTO led to enhancements in the electrical conductivity, the Seebeck coefficient and power factor. It was found that LSTO integrated with 0.6 wt % graphene exhibited the best figure of merit and power factor. The figure of merit was over 0.25 at room temperature. 

## 4. Conclusions

In this paper, we presented a review of the electronic and thermal properties of graphene and its up-to-date applications. Graphene has exceptional physical characteristics compared to other usual metals and semiconductors and has the potential for extensive applications. The ambipolar electric field effect in SLG proves that charge carriers have higher mobilities than those of semiconductors. Additionally, the QHE, the unique phenomenon observed with LL, is explained by graphs and unique geometrical phases. Using the experimental results of graphene crystal samples, the visual transparency of graphene was characterized by the fine-structure constant α. In addition to its electronic features, the superior thermal properties of graphene were reviewed. The specific heats of graphene and graphite were derived, and the thermal conductivity of carbon allotropes containing graphene was explained. Graphene has high thermoelectric power, which suggests its potential for use in energy-harvesting applications. These remarkable properties of graphene can be incorporated into versatile systems in various application fields: optical devices, electronic and thermal sensors and energy management systems. Here, state-of-the-art examples based on graphene were presented. Despite the development of many graphene-based electronic systems with unique properties, challenges and problems related to the development of materials for commercial electronics still exist and need to be studied further. By resolving such challenges and problems, graphene will be in the limelight as an active material for future electronics.

## Figures and Tables

**Figure 1 nanomaterials-09-00374-f001:**
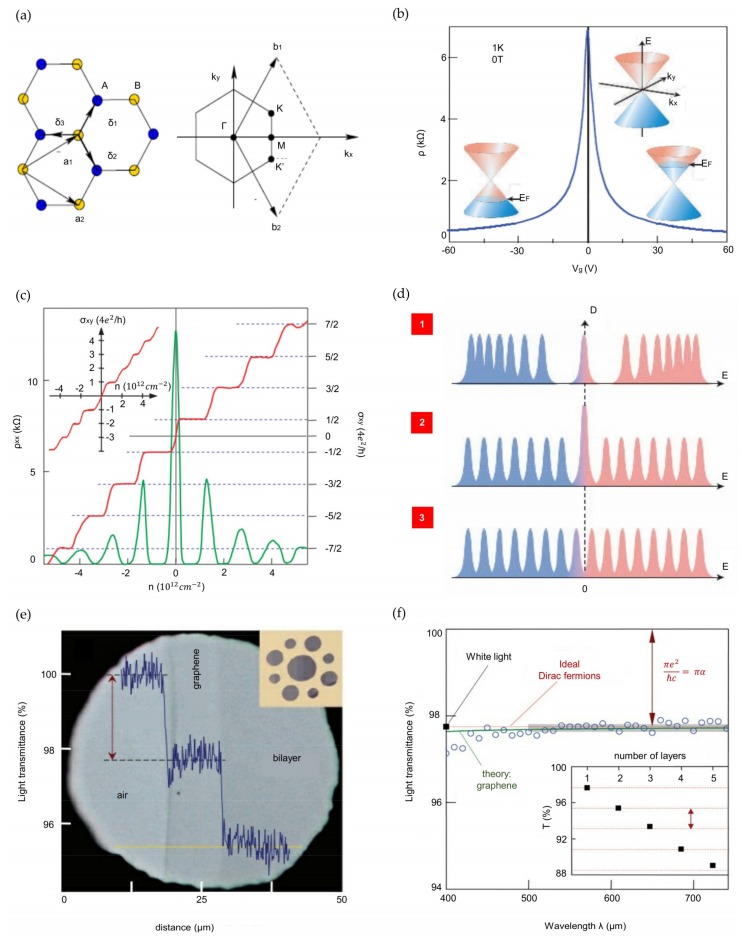
Electronic and optical properties of graphene, (**a**) Hexagonal lattice structure (left) and its Brillouin zone (right); (**b**) Graph of the ambipolar electric field effect of single-layer graphene. The inset indicates the change in Fermi energy E_F_ in response to the changing gate voltage *V_g_* in the conical low-energy spectrum *E*(k); (**c**) Hall conductivity σ*_xy_* (red) and longitudinal resistivity ρ*_xx_* (green) of graphene versus carrier concentration. The inset shows σ*_xy_* in double-layer graphene; (**d**) Three examples of Landau quantization in graphene. *D* is density of states; (**e**) Image of a 50-μm diameter aperture covered partly by graphene and its bilayer. (Inset) The designed sample; (**f**) The open circles show the transmittance spectrum of single-layer graphene. (Inset) Black squares indicate the transmittance of white light versus the number of graphene layers; (a) Reproduced with permission from [13]. Copyright Reviews of Modern Physics, 2009. (b), (d) Reproduced with permission from [14]. Copyright Nature Materials, 2007. (c) Reproduced with permission from [7]. Copyright Nature, 2005. (e), (f) Reproduced with permission from [20]. Copyright Science, 2008.

**Figure 2 nanomaterials-09-00374-f002:**
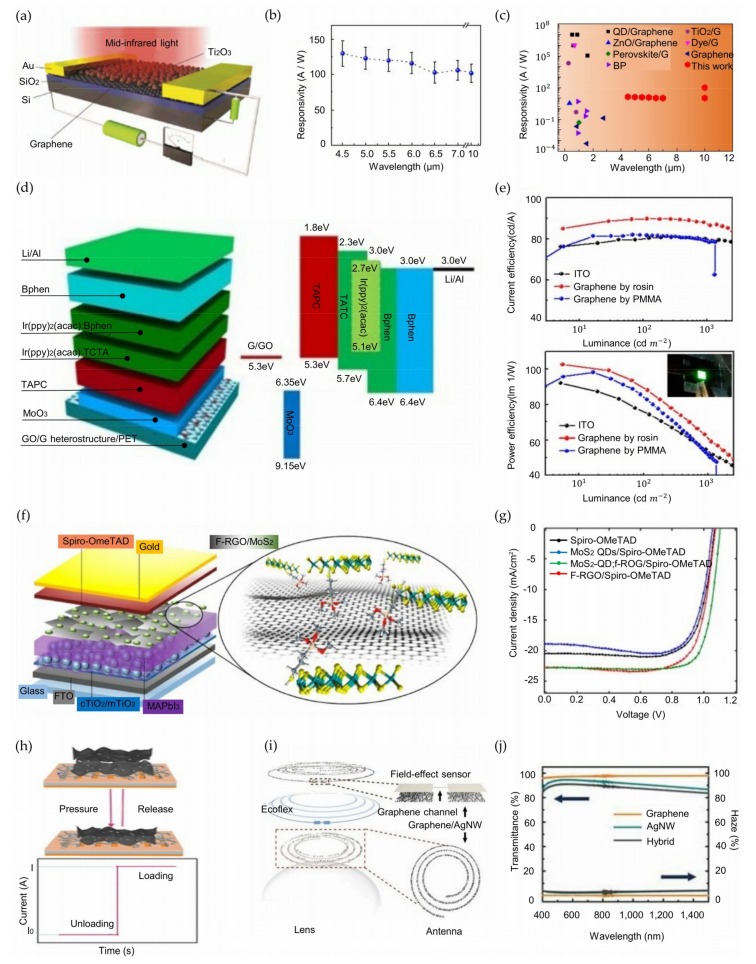
Electronic applications of graphene (**a**) Schematic showing a hybrid graphene/Ti2O3 photodetector for mid-infrared detection; (**b**) High photodetector responsivity when illuminated with wavelengths of light from 4.5 to 10 μm; (**c**) Comparison of photodetector responsivity with the results of other recent studies; (**d**) Structure of a graphene OLED (left) and its energy-level diagram (right); (**e**) Current efficiency (top) and power efficiency (bottom) versus luminance characteristics of three types of OLEDs. The inset indicates a flexible green OLED with a rosin-transferred graphene anode; (**f**) Schematic of a mesoscopic MAPbI3-based perovskite solar cell; (**g**) I-V curves of measured perovskite solar cells exploiting MoS2 QDs, f-RGO, and MoS2 QDs:f-RGO as an ABL between spiro-OMeTAD and MAPbI3; (**h**) Sensing mechanism of a piezoresistive pressure sensor and current response to loading and unloading. (**i**) Image of multifunctional sensors on a soft contact lens; (**j**) Transmittance and haze of graphene, AgNW films, and graphene/AgNW hybrid structures. (a), (b), (c) Reproduced with permission from [48]. Copyright Nature Communications, 2018. (d), (e) Reproduced with permission from [21]. Copyright Nature Communications, 2017. (f), (g) Reproduced with permission from [70]. Copyright ACS Nano, 2018. (h) Reproduced with permission from [71]. Copyright Small, 2018. (i), (j) Reproduced with permission from [72]. Copyright Nature Communications, 2017.

**Figure 3 nanomaterials-09-00374-f003:**
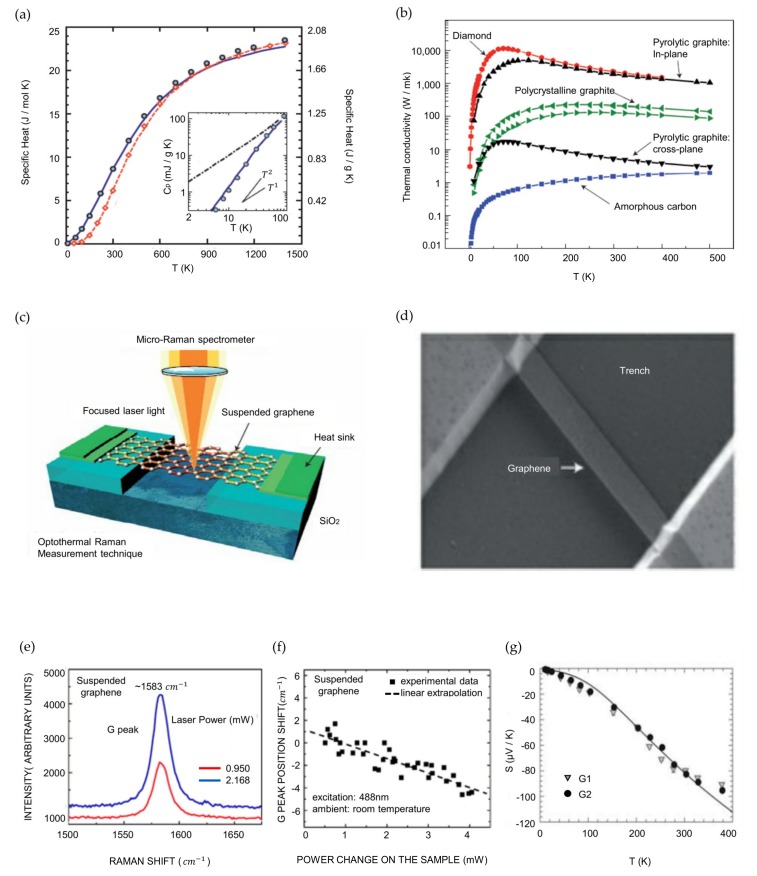
Thermal properties of graphene (**a**) Specific heats of graphite, diamond, and graphene. The inset compares the specific heats of graphene and graphite at low temperature; (**b**) Thermal conductivity of bulk carbon allotropes as a function of temperature; (**c**) Figure of a graphene layer suspended across a trench and the measurement of the thermal conductivity; (**d**) SEM image of a suspended graphene layer across a 3-μm-wide trench in a Si wafer; (**e**) Raman spectrum of graphene showing the G-peak region measured at two different power levels; (**f**) Graph showing the G-peak position shift versus power change; (**g**) Graph of measured Seebeck coefficients as a function of temperature for graphene samples. (a) Reproduced with permission from [171]. Copyright MRS Bulletin, 2012. (b), (c), (d) Reproduced with permission from [16]. Copyright Nature Materials, 2011. (e), (f) [176]. Copyright Nano Letters, 2008. (g) Reproduced with permission from [18]. Copyright Science, 2010.

**Figure 4 nanomaterials-09-00374-f004:**
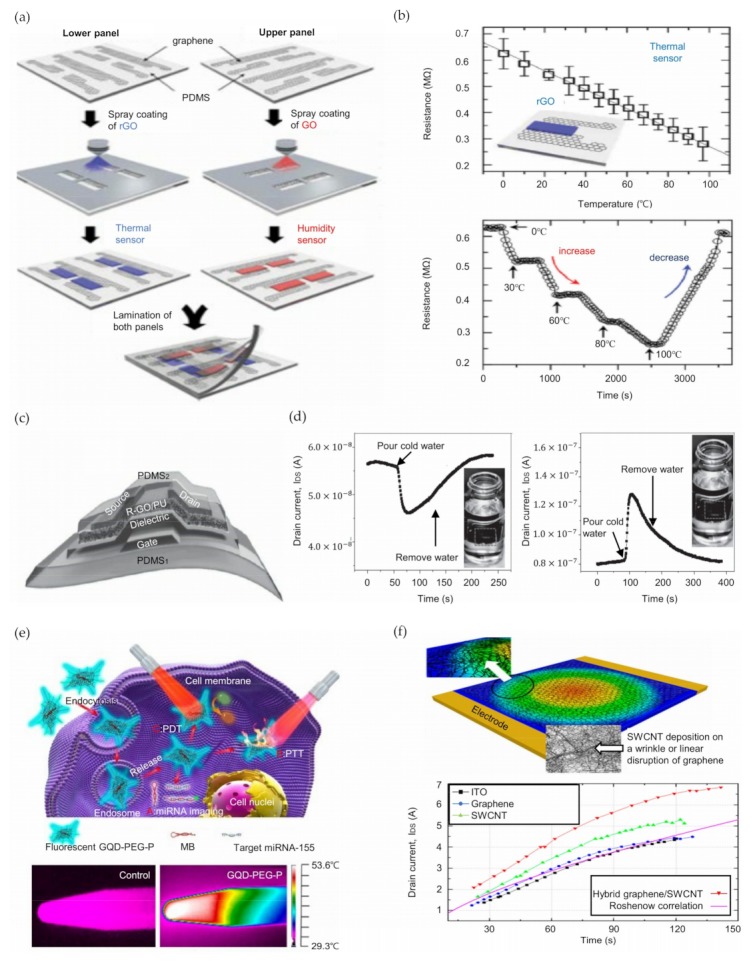
Thermal applications of graphene. (**a**) Schematic of the fabrication process of an all-graphene electronic skin sensing matrix; (**b**) Graph showing resistance vs temperature of an rGO-based thermal sensor (top) and real-time temperature sensing results (bottom); (**c**) Schematic of a device with a transparent and stretchable gate; (**d**) Drain current responses of a TS-gated sensor according to low-temperature water (left) and high-temperature water (right); (**e**) Schematic of a GO-PEG-P theragnostic platform (top) and thermal images of vials with water and GQD-PEG-P solution (100 μg/mL) (bottom); (**f**) Hybrid graphene/SWCNT film deposited on an ITO surface showing high effusivity (top) and heat transfer coefficients on various heating surfaces (bottom). (a), (b) Reproduced with permission from [195]. Copyright Advanced Materials, 2016. (c), (d) Reproduced with permission from [197]. Copyright Advanced Materials, 2016. (e) Reproduced with permission from [199]. Copyright ACS Appl. Mater. Interfaces, 2017. (f) Reproduced with permission from [200]. Copyright Nano Letters, 2016.

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
