# Peer review of "Electronic and Thermal Properties of Graphene and Recent Advances in Graphene Based Electronics Applications"

_nanomaterials, 2019, doi:10.3390/nano9030374_

Reviewer 1 Report

This review paper presented electronic and thermal properties of graphene related to applications. The text is well written and can be accepted if the authors consider the following points which are listed (more or less) in the order they appear in the manuscript. The main objections, except some points where text must be rephrased or some references were missing, is that one to two small paragraphs/parts are lacking. Raman spectroscopy is only mentioned one time whereas it gives valuable information in both electronic and thermal aspects investigated in this paper. The second is that some limitations (wrinkles, crystallite size,…), already listed in the text, may be resumed in a part just before the conclusion, written as a discussion. Moreover, as many acronyms are used in this manuscript, I strongly suggest adding an acronym list at the end of the paper in order to better read this paper. The word review should be included in the title: it will help the reader to find the paper when searching for some bibliography.

Page 1/abstract and introduction

 Line 11: “studied” may be more appropriate than “researched”.

Line 15: Don’t you want to say “review” instead of “introduce”?

Line 11 and 28: “recently” should be avoided as it is not so recent, and this idea is relative. A date of discovery mentioned is preferable.

Line 40: this is a comment that may lead to a little change: I do not know if we can say that “Dirac Fermion” is an electric property. The electronic property is “electrons behaving as massless particles”, and we call that “Dirac Fermion”.

 I think that, as it is a review paper composed of many subparts, a summary should be added. It will better help researchers who are interested only in one of the aspects explained in this work to access more efficiently the relevant information.

Page 3/ Electronic properties

Figure 1: caption corresponding to subplot (a)à is “reciprocal space” more appropriate than “Brillouin zone” here as the vectors b1 and 2 fall outside the Brillouin zone?

Figure 1: subplot (d)à it could be more understandable to write something helping to understand what the z axis is. Moreover: this subplot gives three examples of quantization. Instead of putting “1”, “2”, and “3”, it could be interesting to put some quantum number plus the index 1, 2 or 3, or finding a way to lead to the information: these are “three examples of different quantified states” represented here.

line 86: I would have started a new paragraph with the sentence “The QHE is another factor…”. Moreover, from end of line 86 to line 89, the authors write something about QHE whereas QHE is detailed in another section, at line 90. The relative position should be adapted. QHE has to be explained in the text as well.

From line 97 to 99, the text is not clear. Nu does not appear in any formula. Then: why mentioning nu=+1/2 or -1/2? Something should be changed in the way to formulate the sentence.

Subplot (c)à data concern graphene and bi-layer graphene whereas the texts (line 100 mention “graphite with two layers”; I think the way to call that is bi-layer graphene and not graphite with two layers. Whatever the choice you will do, could you homogenize?

Line 111: I am not sure the word “explained” is the one that express the best the idea. Is “dependent” better suited for the idea you want to highlight?

Line 128: what is the unit of G here? Please complete/modify.

Page 6/Electronic applications:

Line 161: “infrared regime”. Could you be more precise?

Line 165 “schematic” has to be changed by “scheme”

line 190: “decide” is it used correctly? I guess the idea below that sentence is “important factors that inform about the performance”. If yes, please modify. If no, I do not understand the causal link.

Line 207: is the use of “problem” correct? If by problem you mean “all what is written the sentences before”, I suggest finding another word.

Line 251: PSC: I do not find the signification of the acronym

Line 325: I suggest creating a new part. The “electronic sensor part” is too important compared to others, and the part from line 325 to 351 can be moved to a “Electronic sensors for medecine” or something like that.

 In the gas molecule sensors (line 396) I suggest adding the reference: DavidoikJ, D. et al, 2D materials 5 (2018) that deals with graphene gas pumps.

Page 13/ Thermal properties and application part

Line 446: I am not sure about the meaning of “is contributed”. Is “is composed” better suited to explain the idea?

Close to line 446: nothing is said about the electron/phonon coupling. Is its role completely negligible compared to other contributions (like the coupling between graphite layers, as mentioned line 451)?

Raman spectroscopy is mentioned to better having access to thermal properties, but Raman spectroscopy itself is not enough detailed in this manuscript whereas it is a fundamental technique for graphene analysis (with some reference papers cited thousands of times, i.e. the work of A. C. Ferrari). I suggest this recent review on the subject: Merlen A. et al, Coatings 7 (2017) 153 that can also be used as a guide to build a paragraph dedicated to the technique. Building a paragraph on the technique adapted for graphene is not required but suggested as this technique helps to feed the thermal par of the manuscript. From my point of view, Raman spectroscopy can also help in determining how stacked are graphene layers by means of the 2D band (which rises because of a resonance mechanism involving electronic structure), allow to monitor doping by metals or other (Beams R. et al, J. Phys. Condens. Matter 27 (2015)). It can be used to detect traces of molecules (see GERS effect: graphene enhanced Raman spectroscopy, Xu, W. G. et al small 9(2013) 1206 but not limited to this paper). Then, this technique can also be mentioned in the “electronic properties” part of this manuscript. At that step, I do not know if adding a paragraph is necessary or mentioning the good references at the good location in the text is enough.

Line 502: n0 signification has to be written close to the equation.

Line 503: Seebeck coefficient has to be written (i.e S=dV/dT)

Line 506-7: please, give two words about what is the ZT coefficient, and the fact it is dimensionless.

Line 509: K=Kph+Kel has not been written before. I suggest replacing K by Kph+Kel.

One question: ZT has been redefined recently (H. S. Kim et al PNAS 112 (2015) 8205-8210). Does it affect your message in your review? It could be nice, if so, to say two words in relation with that paper.

This topical review (P. Dollfus et al J. Phys. Condens. Mater 27 (2015) 133204) should be incorporated in the bibliography and discussed as it deals with thermoelectric effects of graphene and related nanostructures. Including a line on nanopatterning of nanoribbons could increase the content of this work.

Line 528: to give the reader the idea of how close to market is the development of Ho et al, it could be interesting to mention the size of the sensor matrix developed.

Line 624: the use of “transport” close to the word “heat” obliged me to read twice the sentence, the first reading inducing a wrong interpretation of the sentence. This sentence may be reformulated.

This is an open question: don’t you think a separated additional part at the end of the paper related on the limiting trends of graphene (wrinkle, defects, grain size behavior, electron-phonon coupling or strong e-e interactions,…) could be helpful for the reader to identify rapidly where improvements can be done in the field or what are the limiting things. May be that part could be concluded positively as some unthinkable properties car emerge from that (for example default in the stacking of bilayer graphene with a given magic angle can lead to the superconductivity, Cao Y. et al Nature 556 (2018) 43, which was not mentioned in your manuscript but may be mentioned somewhere). This hypothetical last paragraph (or discussion part that I think is missing) can just sum up some of the points existing in the different parts. I think this suggestion can enhance the attractivity of the paper.

Page 20/conclusion

I do not think that sentence is correct: “… is explained by graphs and unique geometrical phases”. What do you mean by “unique”? Never reported elsewhere? What is the purpose of mentioning that you use graphs to explain things. This is a common thing in papers, I guess. So, I think I missed your idea. Then line 698-699 has to be rewritten to be more understandable.

 Line 710: limelight in one word.

Author Response

The reviewer 1 commented that this review paper presented electronic and thermal properties of graphene related to applications and the text is well written and can be accepted if the authors consider the few points and revise them. We removed the existing improper references and added new references that the reviewer recommended and we found. The reviewer also noted the need for more detail of Raman spectroscopy, some limitations. Therefore, we add the latest and detail research on graphene's properties. Additionally, the reviewer recommended including an acronym list, so we added the acronym list  in the manuscript. We thank that reviewer 1 detailed comments. 

Reviewer 2 Report

The authors present a well organised review article about 

Electronic and Thermal Properties of Graphene and  Recent Advances in Graphene Based Electronics Applications. Some minor corrections /additions

1. The introduction needs to be enriched with a paragraph describing the organisation  of this review.

2. References should be enriched with most recent entries after 2015

Author Response

 The reviewer 2 commented that the authors (we) present a well organized review article about Electronic and Thermal Properties of Graphene and Recent Advances in Graphene Based Electronics Applications. And the reviewer pointed out some minor corrections /additions to enrich introduction with a paragraph describing the organization of this review and references with most recent entries after 2015. So, we added more detail descriptions in the introduction section and we added the recent references about the applications. In the graphene properties parts, the principle references are essential, so we can’t replace those.

Reviewer 3 Report

This is a review paper on graphene and its applications. The paper is well written and is highly informative. The authors explain the electronic structure of graphene in simple terms, which adds value to their paper. However, I am surprised that the authors do not mention anything on graphene simulations, especially using Density Functional Theory (DFT), that predict graphene electronic structure, in particular to graphene use on Lithium and Sodium ion batteries (LIB and NIB). The authors must include a short paragraph just before section 2.2 describing computational work on Li and Na adsorbed on graphene and cite the following work below or similar (major revision). 1. Chan, K.T.; Neaton, J.B.; Cohen, M.L. First-Principles Study of Metal Adatom Adsorption on Graphene. Phys. Rev. B 2008, 77, 235430-235412. 2. Dimakis, N.; Valdez, D.; Flor, F.A.; Salgado, A.; Adjibi, K.; Vargas, S.; Saenz, J. Density functional theory calculations on alkali and the alkaline Ca atoms adsorbed on graphene monolayers. Appl. Surf. Sci. 2017, 413, 197-208.3. 4. Nakada, K.; Ishii, A. DFT Calculation for adatom adsorption on Graphene. In Graphene Simulation, Gong, J.R., Ed. InTech Inc: Croatia, 2011; pp. 3-20. 5. Medeiros, P.V.C.; de Brito Mota, F.; Mascarenhas, A.J.S.; de Castilho, C.M.C. Adsorption of Monovalent Metal Atoms on Graphene: A Theoretical Approach. Nanotechnology 2010, 21, 115701-115706. 6. Lee, E.; Persson, K.A. Li Absorption and Intercalation in Single Layer Graphene and Few Layer Graphene by First Principles. Nano Lett. 2012, 12, 4624-4628. 7. Okamoto, Y. Density Functional Theory Calculations of Lithium Adsorption and Insertion to Defect-Free and Defective Graphene. J. Phys. Chem. C 2016, 120, 14009-14014. 8. Yildirim, H.; Kinaci, A.; Zhao, Z.-J.; Chan, M.K.Y.; Greeley, J.P. First-Principles Analysis of Defect-Mediated Li Adsorption on Graphene. ACS Appl. Mater. Interfaces 2014, 6, 21141−21150. 9. Shiota, K.; Kawai, T. Li atom adsorption on graphene with various defects for large-capacity Li ion batteries: First-principles calculations. Jpn. J. Appl. Phys. 2017, 56, 06GE11-03 Clearly the including information on the computational work done on graphene will enhance the author’s work and its broader impacts.

Author Response

  The reviewer 3 commented that the paper is well written and is highly informative and the authors explain the electronic structure of graphene in simple terms, which adds value to their paper. But the reviewer 3 recommended that we must include a short paragraph just before section 2.2 describing computational work on Li and Na adsorbed on graphene and cite the following work below or similar. So, we added manuscript with the references that the reviewer recommended.

Round  2

Reviewer 3 Report

The paper can be now accepted as is.